# Incidence and Risk Factors for Extremity Osteoradionecrosis after Limb-Sparing Surgery and Adjuvant Radiotherapy

**DOI:** 10.3390/cancers15082339

**Published:** 2023-04-17

**Authors:** Yun-Jui Lu, Chun-Chieh Chen, Shih-Heng Chen, Cheng-Hung Lin, Yu-Te Lin, Chih-Hung Lin, Chung-Chen Hsu

**Affiliations:** 1Department of Plastic and Reconstructive Surgery, Chang Gung Memorial Hospital, College of Medicine, Chang Gung University, Taoyuan 333, Taiwan; 2Department of Orthopedic Surgery, Chang Gung Memorial Hospital, College of Medicine, Chang Gung University, Taoyuan 333, Taiwan

**Keywords:** osteoradionecrosis, extremity sarcoma, soft tissue sarcoma, radiation therapy

## Abstract

**Simple Summary:**

This study investigated osteoradionecrosis (ORN), a complication after radiotherapy, in 198 patients with extremity soft tissue sarcoma (STS). The incidence rate of extremity ORN was 3.5%, mostly located in lower extremities. Clinical presentations included chronic ulcers, soft tissue necrosis, sinus discharge, bone nonunion, and pathological fractures. The ORN group had a significantly higher total radiation dose and greater use of intraoperative periosteal stripping. Repeat surgeries and subsequent soft tissue reconstruction or limb amputation were performed as treatments. Careful surveillance should be taken to manage the risk of ORN in extremity STS patients.

**Abstract:**

Osteoradionecrosis (ORN) is a major complication after radiotherapy. Most studies on ORN have focused on patients with mandibular lesions, with few studies including patients with extremity soft tissue sarcoma (STS). We included 198 patients with extremity STS who underwent limb-sparing surgery and adjuvant radiotherapy between 2004 and 2017. The incidence rate of extremity ORN was 3.5% (7/198), with most lesions (6/7) located in the lower extremities. The mean follow-up time was 62 months. Clinical presentations included chronic ulcers, soft tissue necrosis, sinus discharge, bone nonunion, and pathological fractures. Compared with the non-ORN group, the ORN group had a significantly higher total radiation dose (68 Gy vs. 64 Gy, *p* = 0.048) and greater use of intraoperative periosteal stripping (*p* = 0.008). Repeat surgeries and subsequent soft tissue reconstruction or limb amputation were performed as treatments. The risk and management of ORN in patients with extremity STS was ignored previously. Because the disease is complex and affects both clinicians and patients, careful surveillance should be undertaken.

## 1. Introduction

Soft tissue sarcoma (STS) is rare, accounting for approximately 1% of all adult malignancies. Nearly half of all STS occur in the extremities [1]. More than 100 STS subtypes have been identified, and pathological diagnosis is performed in accordance with the World Health Organization (WHO) classification guidelines [2]. Although different types of treatments are available, surgical resection remains the standard treatment for STS. Limb-sparing therapy is attempted first to achieve satisfactory functional outcomes and quality of life.

Radiation therapy (RT) is an adjunctive treatment administered preoperatively or postoperatively to achieve local control and reduce the local recurrence rate of STS, particularly for tumors with a positive margin and high histological grade. New techniques, such as intensity-modulated RT (IMRT) and proton beam therapy, have advantages over older techniques in reducing peripheral tissue damage [3]. Nevertheless, early and late toxicities of RT may occur, with common ones being edema, dermatitis, compromised wound healing, fibrosis, pathological fracture, and secondary malignancy.

Osteoradionecrosis (ORN) is a complication that occurs most frequently in patients with head and neck cancer after radiation exposure. ORN was first described by Ewing in 1926 [4] and is characterized by an irradiated bone that does not heal over 3–6 months, without any evidence of persisting or recurrent tumor [5]. The disease is pernicious, and treatment involves multidisciplinary decisions. Published articles have focused mainly on patients with head and neck cancer, and only a few of them have discussed other regions. To better understand the bone response to radiation doses in extremity STS, we evaluated extremity ORN, as well as its incidence, presentation, treatment, and risk factors.

## 2. Materials and Methods

We conducted a retrospective study between 1 January 2004 and 31 December 2017 in a single medical center. This study was approved by the institutional review board of Chang Gung Medical Foundation. A total of 391 patients with extremity STS were classified using the International Classification of Diseases, Tenth Edition, Clinical Modification codes C47.1, C47.2, C49.1, and C49.2. First, we excluded patients with missing data or who had not undergone surgery. Subsequently, patients who did not receive limb-sparing surgery and adjuvant RT with curative intent were excluded. Finally, we included and analyzed 198 patients. Figure 1 displays the study diagram.

Data on the demographics, tumor characteristics, treatment details, and clinical outcomes of the patients were collected in encrypted form. Tumor staging and grading were coded in accordance with the American Joint Committee on Cancer Staging Manual, Eighth Edition [6], and the French Federation of Cancer Centers Sarcoma Group histological grading system guidelines, respectively. The pathological report was reviewed, and the tumors were classified in accordance with the WHO classification guidelines [7]. The surgical margin was categorized as negative, close (<1 cm), or positive. The dose, fraction, modality, and timespan of RT were recorded.

### 2.1. ORN Identification

The medical records of each patient with extremity STS were reviewed in detail because the initial clinical presentation may be subtle, and diagnostic codes were unreliable. Clinical presentations ranged from pain, chronic ulceration, and fistula formation with purulent discharge to bone exposure and pathological fracture. Several diagnostic images, primarily those obtained using magnetic resonance imaging (MRI), were examined by a radiologist and two clinicians. MRI images were generally obtained using sequences of T1-weighted image (T1 WI), T2-weighted image (T2 WI), short tau inversion recovery (STIR), and T1-weighted image with gadolinium enhancement. The bone marrow may have low signal intensity on T1 WI, heterogeneous intensity on T2 WI, and hyperintensity on STIR images in ORN patients [8,9]. The increased signal intensity of the surrounding tissue in contrast-enhanced T1 WI indicated the inflammatory process or superimposed infection. In advanced disease, cortical defect, bony sequestrum, or pathological fracture would be presented [10].

The type of operative procedure, such as debridement or sequestrectomy, raised suspicion for ORN. The histopathology of ORN is characterized by diffuse bony necrosis with a hypovascular marrow space, nonviable periosteum, and nonexistent osteoclasts [11,12]. Figure 2 demonstrates the pathological findings of one of the ORN patients.

### 2.2. Statistical Analysis

The dataset was first tested for normality using the Kolmogorov–Smirnov or Shapiro–Wilk test. The difference between the groups (ORN and non-ORN) was evaluated using Fisher’s exact test and the Mann–Whitney U test for categorical and continuous variables, respectively. Two-tailed tests were used, and the significance level was set to 0.05. All analyses were performed using SPSS 26 (SPSS Inc., Chicago, IL, USA).

## 3. Results

A total of 198 patients with extremity STS who were treated using limb-sparing surgery and postoperative RT were included. Most patients were middle-aged, with a median age of 51.5 years, and had no considerable chronic illness (Charlson comorbidity index ≤ 3). Of these patients, 113 were men (57.1%), 85 were women (42.9%), and 41 were smokers (20.7%). The primary tumor was more commonly located in the lower extremity than in the upper extremity (155 patients vs. 43 patients), with the thigh being the most commonly affected anatomical region. A total of 174 patients had primary tumor occurrence, and 24 presented with recurrent disease. All patients with tumor relapse had no history of RT, and all patients underwent wide excision of the tumor with limb preservation. Lymphovascular invasion occurred in 31 patients (15.7%) due to locally advanced growth or adjacency, and periosteal stripping was performed in 14 patients (7.1%) during surgery. Soft tissue reconstruction with local flap transfer (4.0%) or microsurgical reconstruction (4.0%) was performed in patients with unfeasible primary closure.

The clinical and tumor characteristics are presented in Table 1. The most common type of tumor histopathology was undifferentiated pleomorphic sarcoma (45 patients, 22.7%) and myxoid liposarcoma (26 patients, 13.1%), followed by synovial sarcoma (25 patients, 12.6%), myxofibrosarcoma (24 patients, 12.1%), leiomyosarcoma (10 patients, 5.1%), and malignant peripheral nerve sheath tumors (10 patients, 5.1%). The R0 tumor resection was achieved in more than half of the surgical group; however, 42.9% of patients with positive surgical margins remained. Adjuvant RT was initiated immediately after the patient recovered and the surgical wounds healed. The total dose and field size of the RT were determined after being reviewed by the tumor board in our institute. Regular surveillance was arranged, and the mean follow-up period was 62 months. Of the 198 patients, 7 (3.5%) had ORN, and the clinical characteristics of each patient with ORN are listed in Table 2. No significant differences in age, sex, body mass index, comorbidity, smoking status, tumor location, tumor dimension, histopathology, and surgical margin were observed between the ORN and non-ORN groups. However, periosteal stripping was significantly higher in the ORN group than in the non-ORN group (*p* = 0.008). Furthermore, the median of the total radiation dose was higher in the ORN group than in the non-ORN group (68.0 Gy vs. 64.0 Gy, *p* = 0.048).

The ORN group consisted of 4 male and 3 female patients, with a mean age of 51 years. Of the 7 necrotic lesions, 6 were in the lower extremity, and 1 was in the left shoulder joint. Tumor differentiation was mainly high grade, and R0 resection was not achieved. The average radiation dose in the ORN group was 67.4 Gy, and the time to ORN onset ranged from 3 to 67 months. Initial manifestations were chronic ulcer in 2 patients, persistent sinus discharge in 3 patients, and periprosthetic fracture and nonunion fracture in 2 patients (Table 2).

### ORN Case Presentations

A 45-year-old man with left lateral thigh myxoid liposarcoma (pT4N0M0, stage IIIB) underwent an operation and postoperative RT with a total radiation dose of 70 Gy. The patient presented with chronic non-healing wounds on the left lateral thigh with persistent sinus and abscess discharge approximately 2 years after the last radiation course. He received wide excision of the necrotic tissue and hyperbaric oxygen (HBO) therapy. However, due to the pathological fracture with persistent sinus discharge and chronic pain, he underwent high above-knee amputation 4 years after treatment (Figure 3).

Figure 4 shows the case of a 48-year-old man with a recurrent right posterior thigh mass and pulmonary metastases. The initial tumor was resected approximately 10 years ago, and the patient received radical excision for the recurrent tumor, which was diagnosed as undifferentiated pleomorphic sarcoma with a close margin. IMRT was performed with a total radiation dose of 60 Gy, and adjuvant chemotherapy was administered concurrently for systemic control. However, 36 months after RT, the surgical site began to break down with progressive soft tissue necrosis and turbid discharge. The patient underwent repeated surgical debridement, sequestrectomy, local flap transfer (four times), and interlocking nail fixation for a pathologically fractured femoral bone in the following years. After the Masquelet-induced membrane technique was performed by an orthopedic surgeon, the bone achieved fair union.

Another patient, a 57-year-old woman with dedifferentiated liposarcoma of the left lower leg (pT4N0M0, stage IIIB), presented with increasing leg size. Initial surgery removed the tumor along with most of the fibula bone. RapidArc RT was administered with a total dose of 70 Gy. However, the patient encountered a traffic accident 3 months after the treatment, resulting in a fracture of the left distal tibial bone that was fixed using an interlocking nail. Despite these efforts, neither the bone nor the wound healed. The patient developed soft tissue infection, sinus discharge, and infected nonunion, and underwent a series of debridement, sequestrectomy, and free tissue transfer to cover wound defects, as well as tibiotalocalcaneal fusion for the nonunion site. The treatment spanned over 3 years and involved 11 surgeries. Unfortunately, the patient ultimately required a below-knee amputation due to relapsed infection episodes.

In this study, extremity ORN stabilized after a series of surgical treatments in four out of seven patients. The cure rate for patients with extremity ORN was approximately 57% (4/7). However, three patients required limb amputation due to uncontrollable disease.

## 4. Discussion

ORN was first described as radiation osteitis by Ewing in 1926 [4]. In ORN, the exposed bone does not heal over 3 months after irradiation, and persistent or recurrent tumors are not evidenced [5]. In 1983, Marx proposed the widely adopted “three H” theory to describe ORN pathophysiology [4]. According to this theory, hypoxia, hypocellularity, and hypovascularization occur in the irradiated tissue, resulting in tissue breakdown and a non-healing wound. In recent decades, clinical investigations on ORNs have mainly focused on patients with head and neck cancers. In one study, the prevalence rate of mandibular ORN was approximately 5–15%; however, the rate varies significantly in different studies [5]. ORN of other body parts, such as the temporal bone, chest wall, and gluteal region, has only been mentioned in a few studies [13,14,15]. Studies have described several risk factors associated with mandibular ORN, including high radiation dose, RT techniques, smoking, dental extraction, alcohol use, poor oral hygiene, and the primary tumor location [5,16,17,18,19,20].

A 3.5% (7/198) incidence rate of ORN was observed among patients with extremity STS. Non-parametric tests revealed two factors associated with ORN development, a high radiation dose (68.0 Gy vs. 64.0 Gy) and intraoperative periosteal stripping. The commonly applied dose of adjuvant RT was 50 Gy on average, followed by a boost of 10–16 Gy to the tumor bed [21]. In our institute, we have, thus far, adopted the following protocol: for patients undergoing primary surgery, 60–64 Gy will be prescribed for negative margins, 66–70 Gy will be prescribed for microscopic residual tumors, and 70–74 Gy for gross residual tumors [22,23]. Theoretically, a higher radiation dose may provide better local tumor control in patients with a positive surgical margin and high tumor burden. However, the use of a boost remains a topic of debate considering the selection factors and potential toxicities [24,25]. Holt et al. reported that higher doses of radiation (60 or 66 Gy) are associated with a higher risk of pathological fracture [26]. Most patients with ORN had poor tumor differentiation and a positive surgical margin. No significant difference in tumor stage, grade, or surgical margin status was observed between the ORN and non-ORN groups.

A relatively high positive margin rate with a relatively low reresection rate was found. When dealing with extremity soft tissue sarcomas, the surgical goal should be to achieve microscopically negative margins to minimize local recurrence rates, if possible [27]. However, in cases where the tumor is located near critical neurovascular structures, bones, or joints, planned resection with a microscopically positive (R1) margin may be necessary. This approach, combined with postoperative radiation therapy, can still result in outcomes similar to those achieved with negative resection margins [28], while minimizing morbidity and maximizing postoperative functions [29].

The association between local tissue trauma and ORN has been repeatedly mentioned in studies on the mandibular area [20]. We believe that the effect of periosteal stripping on the appendicular skeleton is similar to that of local tissue trauma. The procedure thins the cortex and damages the periosteal vascularity, which may increase the risk of bone fracture and local ischemia. The resulting ischemia could further deteriorate the hypoxia—hypocellularity—hypovascularization conditions. It has been mentioned as one of the risk factors for pathological fractures after treatment for soft tissue sarcoma [30]. We also found a similar result in our patient group, although the extent of periosteal stripping was not well-documented.

The clinical symptoms of ORN vary with the anatomical region, and the general presentation includes pain, paresthesia, chronic ulcers, sinus drainage, tissue necrosis, bone exposure, and pathological fractures. Other specific and serious manifestations, such as trismus and orocutaneous fistula due to ORN of the jaw, otorrhea and hearing loss due to ORN of the temporal bone area, brain abscess, and cerebrospinal fluid leakage due to ORN of the skull base, and thoracic viscera exposure and empyema due to ORN of the chest wall, have been reported [5,15,31]. The presentation of patients with extremity ORN is similar to that of patients with ORN of other anatomical regions. Therefore, a proper diagnosis and identification of the possible cause, especially the possibility of tumor recurrence, are crucial. The ORN onset time is between 4 months and 3 years after RT [32]. However, in this study, the patients had a broad range of presentation times (3–67 months).

The presentation of ORN may overlap with other disease entities. For example, postradiation fractures have been reported with an incidence of 1.2–6.4% and with well-established risk factors [33]. These fractures can occur without significant trauma due to changes in the mechanical strength of the bone after irradiation [34]. In our study, eight patients (4.0%) experienced pathological fractures, and five of them were managed as ORN based on the clinical history, imaging, and assessment of soft tissue quality. Patients with ORN tend to have more prominent soft tissue problems. Persistent sinus discharge with secondary infection was also frequently observed in our patient group, which is a hallmark of chronic osteomyelitis [35]. The difference between ORN and chronic osteomyelitis could be challenging in the clinical setting. However, based on the patient’s history of previous radiation therapy and evaluation of pathological tissues [36], we are more inclined to diagnose ORN, as there is a certain discrepancy in the treatment principles between the two conditions.

Unlike the head and neck region, the extremities have a greater amount of soft tissue covering the hard tissue. Although ORN diagnosis is primarily based on clinical manifestations, image analysis plays a critical role in detecting underlying soft tissue change. Different imaging modalities are indicated in the head and neck region, including radiographs, computed tomography, MRI, and positron emission tomography [5]. However, in an extremity evaluation, MRI exhibits superior soft tissue contrast and delineation to other types and, therefore, is the choice of imaging modality [37]. In ORN of the jaw, abnormal marrow signal, cortical destruction, and slight-to-mild irregular enhancement are noted in MRIs [5]. Radiological changes in long bones after RT have been described as muscle atrophy, cortical thinning with associated remodeling, and bone infarcts in a study [38]. In our patients, hypointensity in the bone marrow with mixed signals over the surrounding tissue was frequently observed on T1-weighted images. The hyperintense signal on STIR and T1 WI after contrast enhancement over the periphery reflected persistent inflammation and tissue necrosis.

Complete recovery from mandibular ORN with conservative treatment was less than 20% [39] in the 1990s. Therefore, several adjunctive treatments were proposed and trialed. HBO therapy with high oxygen tension stimulates fibroblast function and increases angiogenesis in the hypoxia—hypocellularity—hypovascularization environment [40]. However, in a recent review and clinical trial on mandibular ORN, convincing results were not obtained [41,42]. Two patients underwent HBO therapy in our series, and the disease persisted. The other emerging treatment involved the combination of oral pentoxifylline, tocopherol, and clodronate. Delanian et al., in a review, discussed the effectiveness of this combination in the treatment of radiation-induced fibrosis [43]. The results of this combination treatment for mandibular ORN appeared promising in a recent case report and systemic review [44,45]. However, only one patient received the combination for a short period in our patient group, and the effect was limited.

Surgery with pathological tissue excision as well as subsequent soft tissue reconstruction (with a local or free flap) are the most promising management for advanced ORN. In patients with mandibular ORN who underwent free flap reconstruction of the jaw, a high treatment success rate and nourishment of the local vascularity by the transferred soft tissue were observed [46]. Adequate debridement to remove all of the devitalized soft tissue and bone is essential for the initial treatment. We would collaborate with an orthopedic surgeon to either achieve rigid bone fixation or apply the induced membrane technique if a segmental bony defect is present after sequestrectomy [47]. Systemic control with intravenous or oral antibiotics was always required for the secondary infection and was also supplemented with antibiotic-containing beads locally. The decision between locoregional tissue transfer or microsurgical reconstruction would be justified based on the patient’s clinical condition and the defect size. Even after all of the efforts, one patient in the ORN group relapsed and ended up with below-knee amputation due to uncontrollable disease.

The limitations of this study include the retrospective approach and relatively small sample size. The progression of the surgical concept and newer radiation modalities may also affect the clinical outcome. Different surgeons may follow different wound management practices, thereby potentially introducing additional bias in this study.

## 5. Conclusions

In this clinical study on extremity ORN, we found that the incidence rate of ORN was 3.5% among patients with extremity STS. Periosteal stripping during the initial operation and a high radiation dose were identified as risk factors for extremity ORN. Compared with mandibular ORN, the presentation of extremity ORN is obscure, making disease management challenging. Treatment options, including HBO, drugs, and surgical excision with soft tissue reconstruction, may be applied in patients with extremity ORN. More clinical studies are warranted for a cautious assessment and early disease prediction.

## Figures and Tables

**Figure 1 cancers-15-02339-f001:**
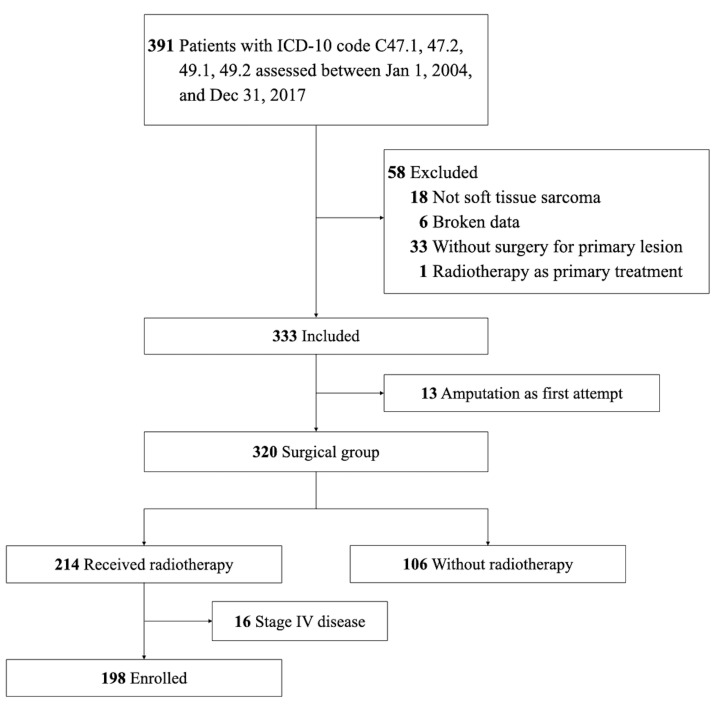
Flowchart of the selection and grouping of patients with extremity STS.

**Figure 2 cancers-15-02339-f002:**
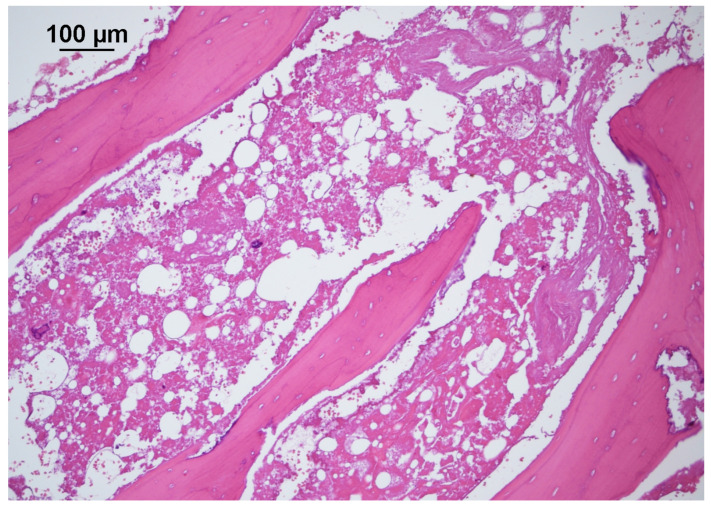
There is extensive necrosis of bony trabeculae and bone marrow tissue. No prominent inflammation is seen.

**Figure 3 cancers-15-02339-f003:**
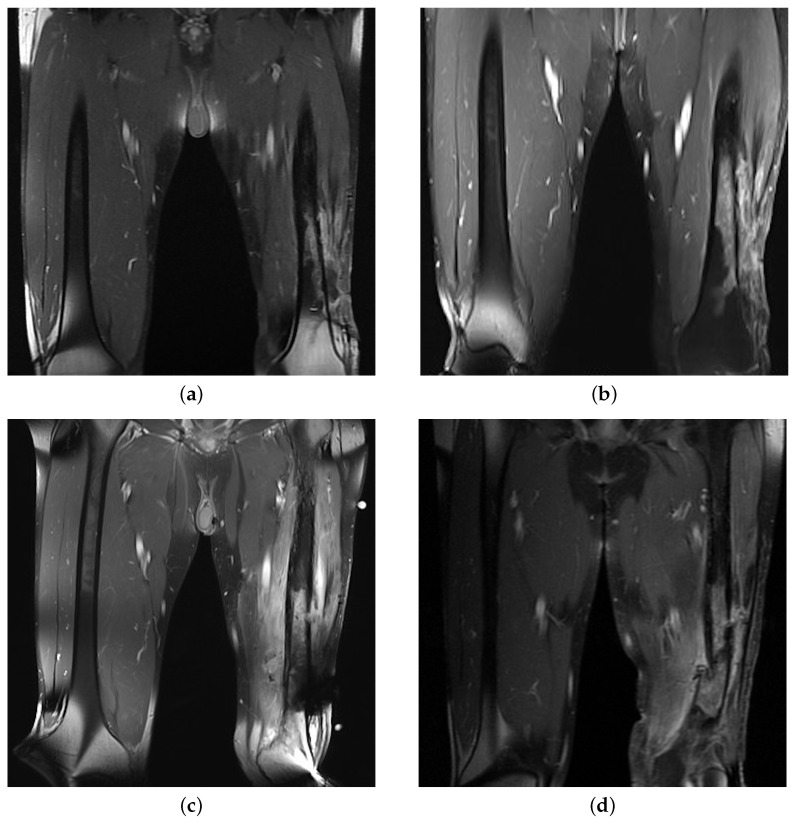
(**a**) Contrast-enhanced T1-weighted image over a 21-month follow-up after RT revealing soft tissue inflammation, bone marrow edema, and skin breakdown with sinus drainage. (**b**) MRI 3.5 years after RT, revealing lesion expansion. (**c**) MRI 4 years after RT, revealing severe tissue inflammation on the medial and lateral thigh, bone marrow edema, and one cortical defect in the left lateral femoral shaft. (**d**) MRI 4.5 years after RT revealing a pathological fracture of the left femoral shaft.

**Figure 4 cancers-15-02339-f004:**
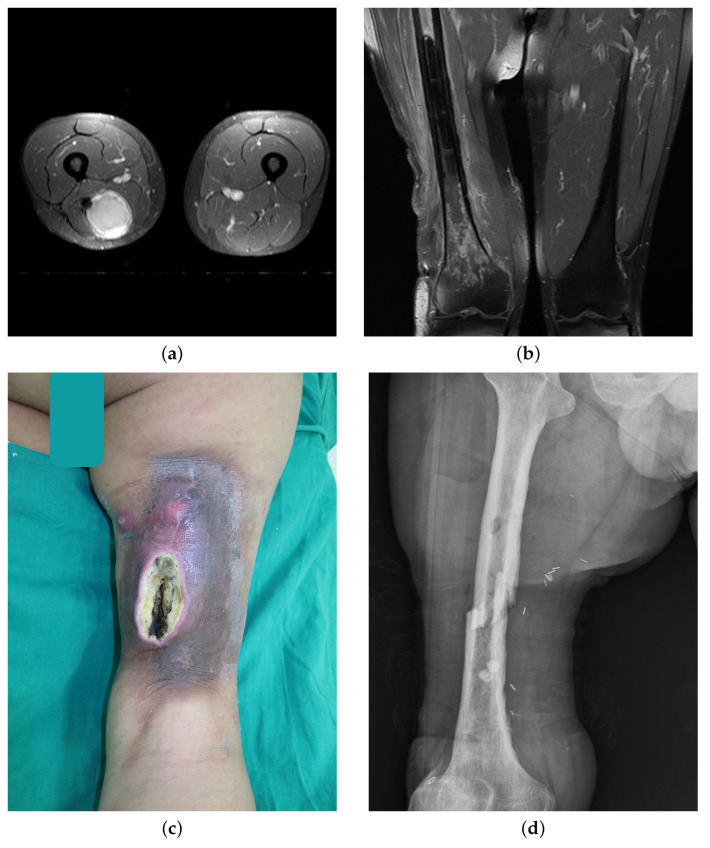
(**a**) Right posterior thigh intramuscular tumor, 4 × 5 × 6.9 cm${}^{3}$. (**b**) Right femur osteonecrosis, fluid collection, and marrow replacement. (5 years after RT); (**c**) Full-thickness soft tissue necrosis with secondary infection and purulent discharge (3.5 years after RT); (**d**) pathological fracture of the right femoral shaft (5.5 years after RT).

**Table 1 cancers-15-02339-t001:** Patient characteristics.

	All	Without ORN	With ORN	*p*
	198	191	7	
Sex				0.654
Male	113 (57.1%)	109 (57.1%)	4 (57.1%)	
Female	85 (42.9%)	82 (42.9%)	3 (42.9%)	
Age, median (IQR)	51.5 (28)	52.0 (28)	51.0 (33)	0.971
BMI, median (IQR)	24.3(5.4)	24.2 (5.3)	27.6 (5.3)	0.121
Smoker	41 (20.7%)	38 (19.9%)	3 (42.8%)	0.157
Alcohol	19 (9.6%)	18 (9.4%)	1 (14.3%)	0.512
Betel nuts	5 (2.5%)	4 (2.1%)	1 (14.3%)	0.166
Hypertension	64 (32.3%)	61 (31.9%)	3 (42.9%)	0.683
CCI				0.301
≤3	167 (84.3%)	162 (84.8%)	5 (71.4%)	
>3	31 (15.7%)	29 (15.2%)	2 (28.6%)	
Tumor				0.203
Primary	174 (87.9%)	169 (88.5%)	5 (71.4%)	
Recurrent	24 (12.1%)	22 (11.5%)	2 (28.6%)	
Location				0.238
Upper limb, proximal	24 (12.1%)	22 (11.5%)	2 (28.6%)	
Upper limb, distal	19 (9.6%)	19 (9.9%)	0	
Lower limb, proximal	107 (54.0%)	105 (54.9%)	2 (28.6%)	
Lower limb, distal	48 (24.2%)	45 (23.6%)	3 (42.9%)	
Depth				0.452
Superficial	47 (23.7%)	46 (24.1%)	1 (14.3%)	
Deep	144 (72.7%)	138 (72.3%)	6 (85.7%)	
Re-resection	30 (15.2%)	29 (15.2)	1 (14.3%)	0.713
Periosteal stripping	14 (7.1%)	11 (5.8%)	3 (42.8%)	0.008 *
Lymphovascular invasion	31 (15.7%)	29 (15.2%)	2 (28.6%)	0.301
Tumor classification				0.644
T1	56 (28.3%)	54 (28.3%)	2 (28.6%)	
T2	77 (38.9%)	74 (38.7%)	3 (42.8%)	
T3	30 (15.2%)	30 (15.7%)	0	
T4	34 (17.2%)	32 (16.8%)	2 (28.6%)	
Histologic grade				0.63
G1	48 (24.2%)	47 (24.6%)	1 (14.3%)	
G2	53 (26.8%)	50 (26.2%)	3 (42.8%)	
G3	86 (43.4%)	83 (43.5%)	3 (42.8%)	
Margin				0.242
Negative	33 (16.7%)	33 (17.3%)	0	
Close	80 (40.4%)	78 (40.8%)	2 (28.6%)	
Positive	85 (42.9%)	80 (41.9%)	5 (71.4%)	
Pathology				0.630
Undifferentiated pleomorphic sarcoma	45 (22.7%)	42 (21.9%)	3 (42.8%)	
Myxoid liposarcoma	26 (13.1%)	25 (13.1%)	1 (14.3%)	
Synovial sarcoma	25 (12.6%)	24 (12.6%)	1 (14.3%)	
Myxofibrosarcoma	24 (12.1%)	24 (12.6%)	0	
Leiomyosarcoma	10 (5.1%)	9 (4.7%)	1 (14.3%)	
MPNST	10 (5.1%)	10 (5.2%)	0	
Other	58 (29.3%)	57 (29.8%)	1 (14.3%)	
Dosage (Gy), median (IQR)	64.0 (6.0)	64.0 (6)	68.0 (6)	0.048 *
Adjuvant chemotherapy	17 (8.6%)	16 (8.4%)	1 (14.3%)	0.472
Repeated RT	16 (8.1%)	13 (6.8%)	3 (42.8%)	0.057

* Statistically significant; BMI, body mass index; CCI, Charlson comorbidity index; IQR, interquartile range; MPNST, malignant peripheral nerve sheath tumor; ORN, osteoradionecrosis.

**Table 2 cancers-15-02339-t002:** Characteristics of patients with extremity ORN.

Case	Sex	Age	Location	Diagnosis	Margin	Radiation Dose (Gy)	Lag Time (m)	Presentation	Management	Outcome
1	M	76	Left lower leg	Undifferentiated pleomorphic sarcoma	Positive	66	3	Chronic ulcers, pathological fracture	Debridement, sequestrectomy, bone graft, ORIF, flap reconstruction	Nonunion bone
2	F	69	Left ankle	Leiomyosarcoma	Positive	60	13	Chronic ulcers, bone exposure	Debridement, flap reconstruction	Healed wound
3	F	26	Left hip	Undifferentiated pleomorphic sarcoma	Positive	74	67	Periprosthetic fracture, persistent sinus discharge	Debridement, hip joint arthroplasty, flap reconstruction	No infection sign
4	M	36	Left upper arm	Synovial sarcoma	Positive	70	14	Persistent sinus discharge, wound poor healing	Debridement, synovectomy	Shoulder disarticulation
5	M	45	Left thigh	Myxoid liposarcoma	Close	70	23	Persistent sinus discharge, wound poor healing, pathological fracture	Debridement, HBO therapy	AK amputation
6	F	57	Left lower leg	Dedifferentiated liposarcoma	Positive	70	3	Nonunion bone	Debridement, bone graft, HBO therapy, ORIF, flap reconstruction	BK amputation
7	M	48	Right thigh	Undifferentiated pleomorphic sarcoma	Close	60	36	Persistent sinus discharge, soft tissue necrosis, nonunion bone	Debridement, sequestrectomy, ORIF, flap reconstruction, pentoxifylline, tocopherol	Bone union, wound healed

AK, above the knee; BK, below the knee; ORIF, open reduction and internal fixation; HBO, hyperbaric oxygen.

## Data Availability

The data are available from the corresponding author upon request.

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
