# Peer review of "Incidence and Risk Factors for Extremity Osteoradionecrosis after Limb-Sparing Surgery and Adjuvant Radiotherapy"

_cancers, 2023, doi:10.3390/cancers15082339_

Round 1

Reviewer 1 Report

In the current study, the authors investigated the appearance and rate of osteoradionecrosis in soft tissue sarcoma patients following wide tumour resection and postoperative radiotherapy. Although they presented clinical presentation of suspected osteoradionecrosis together with treatment options and defined some risk factors, the novelty of the study in comparison to already existing literature is minor. 

First, several studies have been published dealing with exactly the same topic. Second, revision of methods, statistics and results (by at least providing information on distribution of variables, as well as reason to choose one or the other statistical test) is required. Third, a thorough discussion of literature, clearly presenting the potential novelty of the study in comparison to existing ones, is strongly recommended.

Please do see comments given below: 

1. The authors should thoroughly discuss what exactly their study added to literature – there are already several articles dealing with influence of radiotherapy on bone in soft tissue sarcoma patients – some are provided below (none of them has been cited in the current study):

Gortzak Y, Lockwood GA, Mahendra A, et al. Prediction of pathologic fracture risk of the femur after combined modality treatment of soft tissue sarcoma of the thigh. Cancer. 2010;116(6):1553-1559. doi:10.1002/cncr.24949

Bartelstein MK, Yerramilli D, Christ AB, et al. Postradiation Fractures after Combined Modality Treatment in Extremity Soft Tissue Sarcomas. Sarcoma. 2021;2021:8877567. Published 2021 Mar 15. doi:10.1155/2021/8877567

Bretschneider T, Michelitsch C, Frima H, Furrer M, Sommer C. Pathologic femur fractures following surgery and radiotherapy for soft tissue sarcomas: A case series. Int J Surg Case Rep. 2021;84:106062. doi:10.1016/j.ijscr.2021.106062

Bishop AJ, Zagars GK, Allen PK, et al. Treatment-related fractures after combined modality therapy for soft tissue sarcomas of the proximal lower extremity: Can the risk be mitigated?. Pract Radiat Oncol. 2016;6(3):194-200. doi:10.1016/j.prro.2015.09.004

Holt GE, Griffin AM, Pintilie M, et al. Fractures following radiotherapy and limb-salvage surgery for lower extremity soft-tissue sarcomas. A comparison of high-dose and low-dose radiotherapy. J Bone Joint Surg Am. 2005;87(2):315-319. doi:10.2106/JBJS.C.01714

2. Why did the authors decide that “acute and chronic inflammation without tumor recurrence” (page 2/10, line 62) was indicative of osteoradionecrosis? In combination with a fistula, this may rather be indicative of osteomyelitis and should be treated as a separate entity. Please provide more reasons why the herein outlined criteria were used to define osteoradionecrosis rather than any other likely pathology (e.g. chronic wound healing deficit with secondary osteomyelitis).

3. Results: Rather than writing “undifferentiated sarcoma”, the authors should adhere to the generally used terminology of “undifferentiated pleomorphic sarcoma”.

4. For any continuous variables, the authors should provide means with corresponding standard deviations, or medians with respective interquartile ranges. In case of non-normal distribution, the latter one should be used, whereas for normally distributed continuous variables, means and standard deviations would apply. 

5. Related to the comment above: As the authors used Man-Whitney-U-tests to compare continuous variables between groups, it is assumed that some of the variables were non-normally distributed. Their distribution should be checked by applying specific statistical tests, e.g. Shapiro-Wilk test. Furthermore, in case a test for non-normally distributed variables is used, medians and corresponding interquartile ranges have to be provided instead of means. Please correct accordingly throughout the manuscript.

6. The authors conclude that “this is the first clinical study on extremity ORN” which is – given the available literature spanning nearly 20 years – untrue. 

Overall, and related to the comment above, the authors should rather focus on the clinical presentation of osteoradionecrosis in their cohort, as they have well worked-up this information, together with potential treatment options. This may help to improve the significance of the manuscript. In the current form, though, it does not add significant novelty to already existing literature.

Reviewer 2 Report

p><br data-mce-bogus="1">Thanks for asking me to review this manuscript

Please use current WHO reference (not 2013 version)quoting the correct number of STS subtypes.

On what basis >70Gy radiation was used -please quote references

Page 7 Figure C please edit (modesty of pt.who consented for clinical photograph should be respected.

What is the basis of the conclusion 'More clinical studies are warranted for a cautious assessment and early disease prediction'. as >70Gy radiotherapy is a drastic intervention.

It is also interesting to note patients with 'adipocytic tumours' received RT -please clarify what are the adipocytic tumours.

Did any patient receive prep RT? If not, why not?

Please comment about high positive margin rates 42% vs low re-resection rates 15%

How many pathological fractures were seen in the cohort and how many were due to tumour recurrence vs ORN?

Nevertheless, despite unconventional use of high dose RT, the manuscript may be interest to some clinical oncologists

Reviewer 3 Report

- In the M&M section, it is mentioned 113 patient were male and 83 were female, this adds up to 196 and not 198 patients. In table 1, it is mentioned 85 patients were female. This adds up to the 198 patients mentioned eralier. Please correct in the text. 
- Please describe the criteria for ORN on MRI in more detail. Which sequences were used? Did you make use of the unenhanced T1 and fluid sensitive sequences?
- Was the radiologist mentioned in the text a dedicated MSK radiologist? 
- The term 'malignant fibrous histiocytoma' is no longer used, the term nowadays is 'undifferentiated pleomorphic sarcoma'. Please correct.

Round 2

Reviewer 1 Report

The authors have addressed the comments raised. I have no further improvement suggestions.

Author Response

Dear Reviewer,

Thank you for taking the time to review our manuscript and for providing valuable feedback. Your feedback has helped us to refine our work a lot.

Sincerely, 
Yun-Jui Lu, Chung-Chen Hsu

Reviewer 2 Report

Thank you for 'revising' the manuscript

How many subtypes of STS are there according to latest WHO classification? 50?? Please correct.

Please consider previous edits suggested and incorporate them in the manuscript 

-rationale for using >70 GY RT, 

Why re-excision was not considered after positive margins
